# Tracing the function expansion for a primordial protein fold in the era of fold-based function prediction: β-trefoil

**Moushmi Goswami, Subhashini Srinivasan** *

Institute of Bioinformatics and Applied Biotechnology, Bangalore, India

* ssubha@ibab.ac.in

## Abstract

The incredibly narrow protein fold bottleneck, which separates the billions of unique proteins on one side to deliver diverse biological functions on the other, arises from folds that tolerate mutations during evolution. One such fold, called the β-trefoil, is present in functionally diverse proteins including cytokines involved in the immune system such as interleukin-1. The unrecognizable sequence-level diversity, even among paralogs of interleukin-1 within the same chromosomal locus, suggests the resilience of this fold to mutational onslaught. Furthermore, β-trefoil domain containing-proteins are known to coexist with other domains to achieve functional diversity. In this study, we challenge the reach and limitations of function prediction using fold-fold comparison using β-trefoil fold as an example. We identified proteins containing β-trefoil fold belonging to thirty-two distinct functional classes based on diverse domain architecture and/or functional annotation by mining both the PDB and AlphaFold databases using fold-fold comparison. Among the proteins with novel domain architecture we find β-trefoil along with chitinase, lipase, β-glucosidase, protein kinase, peptidoglycan-binding + peptidase matrixin, glycosyl hydrolases family 3 + PA14 + fibronectin type- III, alpha galactosidase A, PhoD-like phosphatase, insecticidal crystal toxin, trypsin, alginate lyase and two novel structurally uncharacterized domains. We demonstrate that fold-fold comparison can extend function prediction beyond the reach of sequence-based approach and provides an opportunity to discover novel domain architecture associated with known folds. However, since extending fold similarity to functional similarity may be challenged by convergent fold evolution, we explore if β-trefoil may be a convergent evolution and share our hypothesis.

## Introduction

Nobel Laureate Francois Jacob, in his essay on Evolution and Tinkering, stated that nature is a better tinkerer than an inventor. As evolution proceeds to create complex organisms from a small repertoire of protein folds, one can expect accumulating

**Data availability statement:** All relevant data are within the paper and its Supporting Information files, including structural-based sequence alignment of all 64 structures representing 32 functions in fasta format.

**Funding:** The author(s) received no specific funding for this work.

**Competing interests:** The aurthors declare no competing interest

**Abbreviation:** PDB, Protein Data Bank; BFD, Big Fantastic Database; DNA, Deoxyribonucleic Acid; EST, Expressed Sequence Tag; BLAST, Basic Local Alignment Search Tool; HMM, Hidden Markov Model; CASP, Critical Assessment of Structure Prediction; ML, Machine Learning; AI, Artificial Intelligence; r.m.s.d., Root Mean Square Deviation; AFdb, AlphaFold database; CATH, Class, Architecture, Topology and Homology; DALI, Distance-matrix ALIgnment; 3Di, Three-Dimensional Interactome; PFam, Protein Families; RefSeq, Reference Sequence; SCOP, Structural Classification of Proteins.

diversity in the amino acid sequences of proteins sharing the same fold. Considering that protein functions are dictated by their unique three-dimensional structures, the accepted mutations during evolution are expected to preserve the overall structural fold of the original protein. While, in some functional families, one can recognize the remnant sequence motifs of the ancient protein, there are protein fold-families with no recognizable sequence signatures, yet sharing folds and, perhaps, function.

In 1992, Cyrus Homi Chothia estimated the number of unique folds to be as low as one thousand [1]. As of 2023, the protein data bank (PDB) has 3D structures of 227,933 proteins with ~7000 clusters less than 30% identity representing only 1,257 unique folds. The five-fold reduction in the number of folds from sequence-based clustering reflects both i) the diversity in sequences within a family below the 30% identity cutoff, and ii) the differing domain architecture within a protein family limiting homology over lengths of the proteins. More recently, the number of sequence clusters in the BFD (big fantastic database), using a collation of 2 billion proteins, is as large as 65,983,866 using a 30% identity cutoff and 90% coverage over the length of proteins [2].

The exponential lag in solving 3D structures of proteins and, until recently, our inability to predict structures of divergent and novel proteins from primary sequences, made sequence homology based-function prediction a norm for the last several decades. We have come a long way since the efforts by Dayhoff in collating thousands of published protein sequences [3]. The development of tools for sequence homology-based function prediction kept pace with growing sequence databases, bridging the lag in structure determination. For example, the early 1990's saw a boom in DNA sequence databases from EST-sequencing efforts, triggering an unprecedented race to discover novel paralogs of genes of therapeutic importance. Function prediction from fragmented DNAs was the need of the time, which was met wholesomely by the timely development of the most popular molecular biology tool in 1990 called BLAST [4]. To improve the sensitivity in sequence-based structure prediction, among divergent paralogs/homologs, hidden markov models (HMM) for individual fold families were developed [5].

Theoretically one should be able to predict functions of divergent proteins using the shared folds. Parallel efforts continued to predict structures of proteins based on homology at the sequence-level. However, the progression of homology-based structure prediction had to wait for decades until incremental progress on many fronts converged to advance structure prediction by deep learning. These advances include the incorporation of multiple templates, the refinement of secondary structure prediction, the optimization of *ab initio* loop modeling, the use of backbone-dependent rotamer libraries for side-chain placement, cooperative substitutions, deluge in protein sequences from next generation sequencing technologies, advances in artificial intelligence/ machine learning and GPU machines [6]. That the native fold is present among the thousands of simulated structures kept the lights-on for researchers through community-based challenge [7]. Wodak et al. has provided a detailed review of progress in methodologies achieved via collective intelligence through community experiments like CASP (critical assessment of structure prediction) and other efforts

[8]. Significant improvements in methods were observed by CASP10 by introducing co-evolution to guide the folding process [9]. Parallel to this effort, the deluge in protein sequences from advances in next generation sequencing revealed more and more coevolving pairs within folding domains adding additional distance constraints between distal amino acids in the primary sequence that are likely to be spatially proximal — advancing structure prediction by leaps and bounds. By 2020, the convergence of collective and artificial intelligence along with advances in high-throughput sequencing technologies, we entered an era when decades happen in weeks. By the time of CASP14, AlphaFold2 predicted 3D structures of test proteins to unprecedented accuracy [10].

Currently, AlphaFold-predicted structure database (AFdb) contains hundreds of millions of predicted structures providing the basis for functional genomics using fold-fold comparison. Methods to predict functions using AFdb are on the rise. In one such effort, using DeepFRI, the contribution of AFdb in enhancing the accuracy of protein function prediction achieved previously by just using PDB was demonstrated [11]. Their results highlighted the importance of AFdb structures among all three Gene Ontology (GO) categories.

The success of AlphaFold in predicting high-quality tertiary structures inspired development of PANDA-3D [12], which is a novel computational tool designed to predict protein functions by leveraging the high-quality 3D structural models in AFdb. Developed as a deep-learning-based method, PANDA-3D focuses on combining the geometric vector perceptron graph neural networks and transformer decoders for multi-label classification of GO over the traditional function prediction tools using sequence-based annotation such as in PANDA [13].

The research by Durairaj et al. explores the diversity of protein structures and functions through advanced computational tools, leveraging the AFdb [14]. By integrating structure, sequence similarity and machine learning tools like DeepFRI, the study developed an annotated sequence similarity network to explore protein functions and evolutionary relationships. The study analyzed over 350 million protein sequences from UniProt, revealing that many proteins remain unannotated/uncharacterized due to limitations in sequence-based annotation methods. Their efforts significantly enhance understanding of these "dark" proteins by the identification of new protein families, such as TumE-TumA, a superfamily of toxin-antitoxin systems and discovery of the β-flower fold, a unique structural motif.

In yet another effort the authors attempted to cluster the 214 million structures in AFdb using a novel structure comparison tool [15]. The authors use Foldseek, a scalable structural-alignment-based method, to identify ~18 million clusters with 2.3 million non-singletons. Out of the non-singleton clusters, 31% was unannotated representing potentially novel structures. Their key findings include species-specific clusters comprising 4% of the total clusters, hinting at *de novo* gene birth. Evolutionary insights suggest most clusters are ancient, with some human immune-related proteins showing remote homology to prokaryotic species, suggesting ancient systems may have been co-opted into specialized functions in higher organisms. Also, they report new domain families, expanding the understanding of protein evolution and function.

Building on these advances, we have used both PDB and AFdb to explore the functional diversity achieved by the primordial β-trefoil fold [16]. Our approach could be considered a bottom-up approach compared to the top-down approach of clustering of all structures in AFdb attempted using Foldseek [17]. Pertaining to existing classification of proteins containing β-trefoil domain, there are 1,544 proteins from acidic fibroblast growth factor and 13 structures with tail fiber receptor-binding protein in the current version of the CATH database [18]. The lack of other functional classification in CATH is both because i) β-trefoil domain is very divergent, missing sequence-based annotation and ii) in multi-domain proteins they constitute less than 50% of the protein. According to SCOP classification, β-trefoil is implicated in eight distinct functional families with capacity to bind to DNA (3brf), to carbohydrate (1jly, 1jlx), to chlorophyll (2dre), to immunoglobulin-like receptor domain (1ira), to actin in a pH dependent fashion (1hcd), proteases (6dwu) and AbfB (1wd3 and 2wd4) [19].

## Results

In this work, the β-trefoil fold was chosen to challenge the reach and limitations of functional classification by fold-fold comparison. β-trefoil fold not only displays functional diversity by promiscuously binding to every class of macromolecules,

but is the most divergent fold family challenging the sequence-based function prediction even among paralogs of the same functional class within the same locus in human [20].

## Functional diversity of β-trefoil fold in PDB

As representative of the β-trefoil fold, we selected accessions for 3D structures of five known proteins including 1i1b, 5uc6, 4p0j, 5bow and 6ncu from PDB. Pairwise alignment of these structures displays greater than 2 Å r.m.s.d. with a sequence identity of less than 30%. During the first iteration, each of these structures were searched for potential homologs against the PDB using DALI [21], a structure comparison tool. The r.m.s.d. cutoff and query coverage for homolog selection was set at 2 Å and greater than 80% respectively to avoid false positives. This process revealed nineteen distinct functions among proteins containing the β-trefoil fold. This is still an underestimate, since we missed other functions present in PDB because of the stringent cutoff. For example, we missed proteins containing the β-trefoil fold in the PDB such as LAG-1 (3brf), AbfB (1wd3) and β-L-arabinopyranosidase (3a21) from our search with a relaxed filtering criteria against AFdb. As shown in Fig 1, we found 51 entries containing β-trefoil domain spanning diverse functions in the PDB including GalNAc, Hemagglutinin, Lectin, Mosquitocidal toxin, Carbohydrate-Binding Family, Fibroblast Growth Factor,

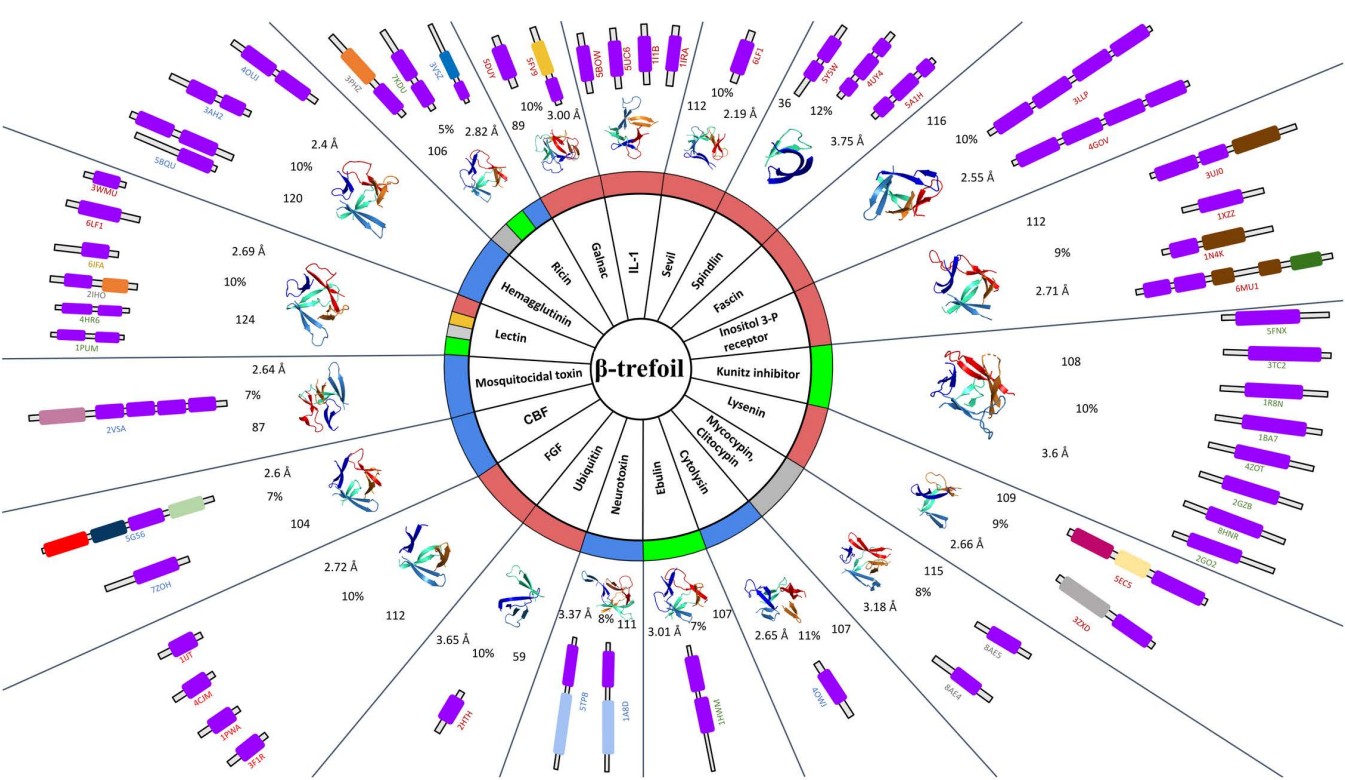

**Fig 1. The core of the sundial represents the β-trefoil fold.** The second ring classifies the proteins into functional families. The third layer shows the distribution of β-trefoil fold in PDB across biological kingdoms, color-coded as blue for bacteria, gray for fungi, green for plants, red for Metazoa, and yellow for protozoa. Fourth layer shows representative β-hairpin without loops from each functional class along with r.m.s.d. and sequence identity compared to interleukin-1 with the PDBid of 5bow. The outermost layer illustrates domain architecture with purple representing the β-trefoil domain and additional domains in unique colors such as Agglutinin (orange), Glycosyl transferase family (dark-yellow), Glycosyl hydrolases family (dark-blue), Clostridium neurotoxin receptor-binding domain (light-blue), RIH domain (brown), Ryanodine receptor-associated domain (dark-green), Unknown domain 1 (light-pink), Unknown domain 2 (dark-pink), Unknown domain 3 (lemon-yellow), Unknown domain 4 (grey), Unknown domain 5 (pastel-green), Spindlin/Ssty family domain (light-green), Cellulase (red), Carbohydrate binding family domain (forest-green).

Ubiquitin, Neurotoxin, Ebulin, Cytolysin, Mycocypin/Clitocypin, Ricin, Lysenin, Kunitz Inhibitor, Inositol-3 Phosphate receptor, Fascin, Spindlin, Sevil and IL-1.

The domain architecture of every protein identified containing the β-trefoil fold from the PDB was characterized using HMM and their domain architecture is shown in the outermost ring within each of the respective functional spokes in Fig 1. Table 1 lists eighteen representative protein structures containing distinct domain architecture for second iteration to search against AFdb. The representative structure for one of the functions, Mosquitocidal toxin, was not taken forward because it did not yield any hits against AFdb. The representative protein structures were queried against the AFdb v2.0 using the 3Di option in Foldseek [17]. During this iteration, the search results were filtered to include only those with an r.m.s.d. below 3 Å and coverage more than 30%. The 3 Å increase in cutoff is based on our experience with searching the PDB during our first iteration, where we missed some functions because of the stringent 2 Å cutoff. The stringency of 30% query coverage for the second iteration was chosen because β-trefoil domain constituted more than 30% of the majority of the query proteins as shown in Table 1 column-2. The cutoff of 30% coverage also allows discovery of novel domain architecture containing β-trefoil domains from the selected query structures without the need for repeating the search only with β-trefoil domain.

Details of the search results from mining AFdb are elaborated in Table 1. The second and third columns contain the query IDs for the representative query structures and their respective kingdoms. The fourth column reports the number of hits with 80% versus 30% coverage from AFdb and the last column highlights the kingdom diversity in AFdb among the 30% hits. With a few exceptions, hits with greater than 80% coverage show less diversity at the kingdom level, as expected. We find considerable kingdom-level diversity from AFdb with respect to the PDB (Fig 1, third ring) for the same functional class. For example, with the query 3vsz from the functional class Ricin, we find AFdb hits in Metazoa, which is missing in PDB. On the other hand, for homologs of ricin from plants is represented in PDB without a single hit from plants in AFdb.

For some of the queries, the number of hits is large for 80% cutoff allowing us to interrogate the accuracy of predicted structures in AFdb with respect to sequence identity. For the eleven query structures with sufficient numbers of hits (bold, Table 1: column 2), the r.m.s.d. versus sequence identity are plotted in Fig 2. As expected, there is a clear inverse correlation between r.m.s.d. and sequence identity among these hits, validating that AlphaFold is a mega homology modeling tool.

Fig 2 shows the plot of r.m.s.d. on the x-axis verses sequence identity on the y-axis for hits covering 80% of the query sequence against AFdb for the eleven query structures inset within the respective plots. The hits from AFdb for the eleven selected query PDB structures shown in Fig 2, vary in pattern widely. A few plots follow a step pattern with each step representing distinct taxonomic classes or orders. For example, each step in the hits from 4gov belonging to Metazoa represent homologs from birds at 78%, to fish at 66%, whale at 55% and Arthropoda at 42% identity. Similarly, hits from 5a1h, from Metazoa, have a step at 78% representing birds and bats. Other predominant patterns in Fig 2 include those with missing structures above ~60% sequence identity and below ~1.5 r.m.s.d cutoffs, such as hits from 4hr6, 2iho and 5tpb. The query 4hr6 is a lectin from snake gourd, belonging to Cucurbitales, with hits between 60% − 50% belong to the same order, hits between 50% − 40% mostly belong to the order Dipsacales and Malpighiales with a few from Rosales and Fabales, and hits between 40% − 30% mostly belong to the order Malvales. The query 2iho is a lectin, which is from mushroom, and finds hits that are also from mushrooms (Basidiomycetes) with a few exceptions from Ascomycetes, which is a sac fungus. The query 5tpb is a toxin from bacterium *Clostridium botulinum* with most hits belonging to the same species and a few from the same genus. Also, in these examples, to explain the gap in the region spanning low r.m.s.d. (<1.5 Å) and high sequence identity (> 60%) we blasted the sequences of the query protein against NR (S1, 1c, 2c and 3c Figs in S1 File) and UniProt databases (S1, 1b and 3b Figs in S1 File) and find that this is resulting from incompleteness in AFdb resulting from missing UniProt entries.

The third pattern is a sudden drop to 60% identity still showing low r.m.s.d. such as in 7kdu, a Rosid and in 1hwm, an Asterid. The 3–4 layers of hits against 7kdu are all plants from different orders. For example, 98% identity is for another Rosid, 66% for plant species under magnolids, and species under monocots show up at 43% identity. Similarly, hits against 1hwm at 81% come under Asterids and those at 43% belong to Rosids.

## Functional annotation of hits from AFdb

Functional annotations of UniProt proteins for all hits from AFdb are by homology at the sequence level. Hence, the annotation in UniProt could be based on partial domain-level homology and may miss domain architecture-based functional annotation. Hereafter, even for hits that are characterized entries in UniProt, independent annotation was attempted using fold-fold comparison. In order to discover novel domain architecture containing the β-trefoil fold, we interrogated hits with query coverage greater than 30% for multi-domain query structures. Kingdom-level diversity in functions for the queries in Table 1 suggests that the fold-fold comparison strategy has the reach to identify homologs across kingdoms. For example, the CBF (carbohydrate binding family) homologs extend to metazoans and is not limited to microbes as observed from PDB.

In Fig 3, we illustrate the domain architecture for the thirteen novel proteins from AFdb containing a β-trefoil domain along with an additional domain, which are not yet represented in the PDB. These proteins were categorized as follows. i) Category-1: Proteins that share their β-trefoil fold, but the structure of the other domain is not yet experimentally determined, ii) Category-2: Proteins whose additional domain is found in the PDB without the β-trefoil fold. In these cases, it was verified that the β-trefoil domain is absent even in the full-length protein of the corresponding PDB structures, indicating that while the additional domain is well represented, its association with the β-trefoil fold is unique to the proteins identified. iii) Category-3: Proteins that are represented in the PDB with both the β-trefoil fold and the extra domain, perhaps missed during our first iteration because of stringent cutoffs.

Among the 13 hits from AFdb, we found four category-1 and eight category-2 hits as shown in Fig 3 and listed in S1 Table in S1 File. Figs 4 and 5 show structures for hits characterized and uncharacterized in UniProt respectively. Figs 4A, 4C, 4E, 4F and 5C-F

**Table 1. Results of AFdb search using 18 representative query structures.**

| Protein function | Representative Query PDB ID | Query Kingdom | Total number of Hits (>80%/30% coverage) | Percentage hits from kingdoms other than query with less than 30% coverage | | | | |
|---|---|---|---|---|---|---|---|---|
| | | | | Bacteria | Fungi | Metazoa | Plant | Protozoa |
| Ricin | **7KDU (47%)** | Plantae | 221/2250 | 95% | 0.04% | 4.3% | None | None |
| Ricin | **3VSZ (31%)** | Bacteria | 2/368 | 20% | 40% | 20% | 20% | None |
| Ricin | **3PHZ (52%)** | Fungi | 30/2038 | 98.93% | None | 0.9% | 0.05% | 0.1% |
| Hemagglutinin | 3AH2 (48%) | Bacteria | 14/721 | 0.4% | 86.5% | 10% | 2% | 3% |
| Lectin | **2IHO (60%)** | Fungi | 37/2271 | 96.8% | None | 2.1% | 0.4% | 0.5% |
| Lectin | **4HR6 (51%)** | Plantae | 152/553 | 98.9% | None | 0.7% | None | 0.2% |
| Carbohydrate binding family (CBF) | 5G56 (20%) | Bacteria | 5/159 | None | 2% | 97% | None | None |
| Neurotoxin | **5TPB (35%)** | Bacteria | 39/42 | None | None | None | None | None |
| Ebulin | **1HWM (51%)** | Plantae | 175/622 | 77.2% | None | 22.8% | None | None |
| Lysenin | 3ZXD (45%) | Metazoa | 3/5 | 50% | 50% | None | None | None |
| Lysenin | 5EC5 (49%) | Metazoa | 1/12 | 100% | None | None | None | None |
| Kunitz inhibitor | **1BA7 (100%)** | Plantae | 631/634 | 100% | None | None | None | None |
| Inositol 3-P receptor | 1N4K (55%) | Metazoa | 6/34 | 100% | None | None | None | None |
| Fascin | **4GOV (31%)** | Metazoa | 1049/1203 | 6.2% | 68.75% | None | 25% | None |
| Spindlin | 5Y5W (30%) | Metazoa | 16/55 | 100% | None | None | None | None |
| Spindlin | **4UY4 (36%)** | Metazoa | 76/188 | 100% | None | None | None | None |
| Spindlin | **5A1H (31%)** | Metazoa | 1069/1223 | 77.7% | 5.5% | None | None | 16.6% |
| Galnac | **5FV9 (27%)** | Metazoa | 102/104 | None | None | None | None | None |

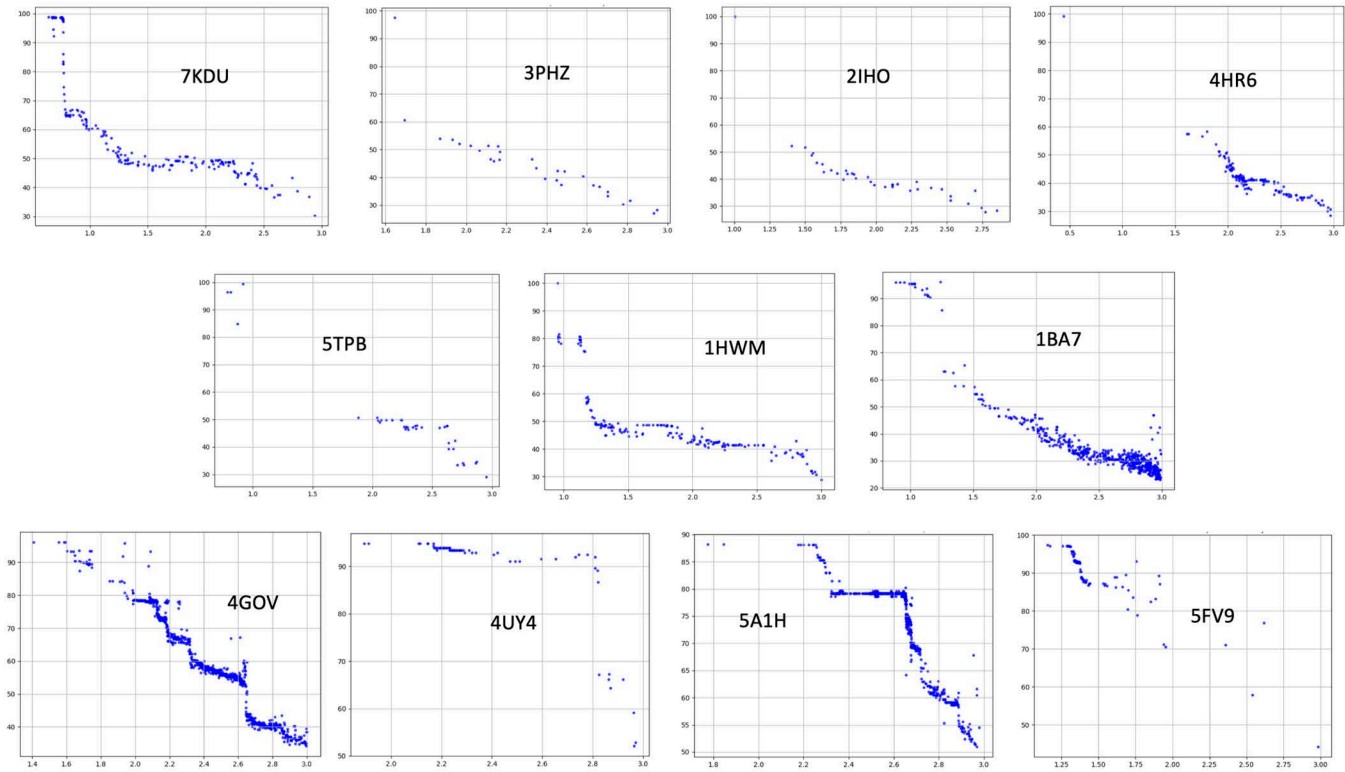

**Fig 2. Each plot shows r.m.s.d. (x-axis: 0-3 Å) versus percent identities (y-axis: 30-100) for the eleven selected queries (bold in Table 1 column 2) against AFdb hits with more than 80% query coverage.**

shows hits belonging to category-2. Examples of Category 2 are i) A0A6I5GSK5, which is annotated as chitinase sharing its chitinase domain with 2dsk (r.m.s.d 1.342 Å), which lacks β-trefoil domain even in the corresponding full-length protein, WP_317259669.1, with 742 amino acids, ii) A0A4Q2KX23, sharing its β-glucosidase domain with 4i3g (r.m.s.d 1.016 Å) lacks β-trefoil domain even in the corresponding full-length protein, WP_230883105.1, with 832 amino acids, iii) A0A3D9WMQ9 shares its protein kinase domain with 3f61 (1.95 Å), which lacks β-trefoil domain even in the corresponding full-length protein, WP_057333683.1, with 626 amino acids, iv) A0A820M8C8 shares its metallopeptidase domain with 1slm (0.824 Å) lacking β-trefoil domain even in the corresponding full-length protein, NP_002413.1, with 477 amino acids, v) A0A191US67 shares its β -L- arabinofuranosidase domain with 4qjy (2.83 Å) lacking β-trefoil domain even in the corresponding full-length protein, WP_371878443.1, with 642 amino acids, vi) A0A844M7R0 shares its phoD domain with 2yeq (0.741 Å) lacking β-trefoil domain even in the corresponding full-length protein, WP_009969242.1, with 583 amino acids, vii) A0A820T4Y8 shares the trypsin domain with 7n7x (0.628 Å) lacking β-trefoil domain even in the corresponding full-length protein, XP_054205941.1, with 436 amino acids, viii) A0A3A9W2R1 shares the alginate lyase domain with 3zpy (0.976 Å) lacking β-trefoil domain even in the corresponding full-length protein, WP_303396801.1, with 446 amino acids. An enlarged view of all proteins in Figs 4 and 5 are shown individually in stereo figures in S2a-f and S3a-g Figs in S1 File.

Figs 4B, 4D and 5A, 5B show category-1 hits. The β-trefoil domains of A0A318NTU0, A0A1I6E245, A0A3L7BAS7 and A0A0U3MKC8 gives an r.m.s.d. of 0.795 Å, 1.867 Å, 0.686 Å and 0.888 Å against a representative β-trefoil domain to confirm the presence of this domain in these novel multi-domain hits from AFdb. For the additional domains of these proteins there is no representation in the PDB, suggesting absence of experimentally determined structure for these domains. Perhaps the annotation for proteins shown in Fig 4B and 4D in UniProt, despite lacking experimental structures in PDB, is based on the presence of the annotated domains in PFam.

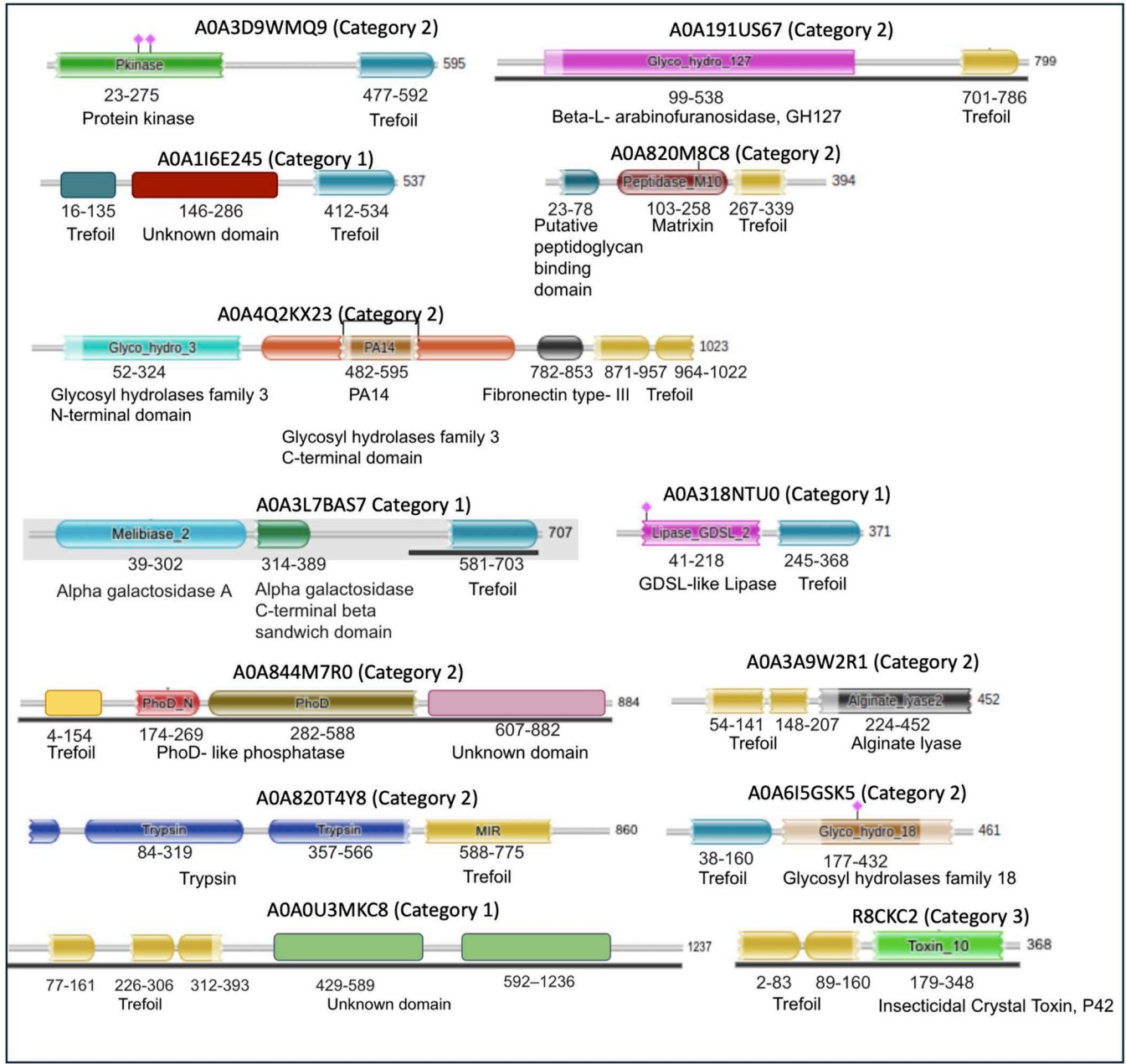

**Fig 3. Novel domain architecture of UniProt proteins containing β-trefoil domains predicted by AlphaFold that are not yet represented in the PDB.** A tabular format is included in S1 Table in S1 File.

## Interrogation of mutation tolerance in β-trefoil

We have produced structure-based alignment for β-trefoil domain of the 51 hits from PDB shown in the outer most layer in Fig 1 and 13 structures from AFdb hits. This alignment was filtered to include only those positions with less than 10% gaps (S2 File). A phylogenetic tree created using this filtered alignment, shown in S5 Fig in S1 File, cluster by kingdom. The same alignment is also used to compute Shannon entropy at each position shown in Fig 6 (bottom), which reveals

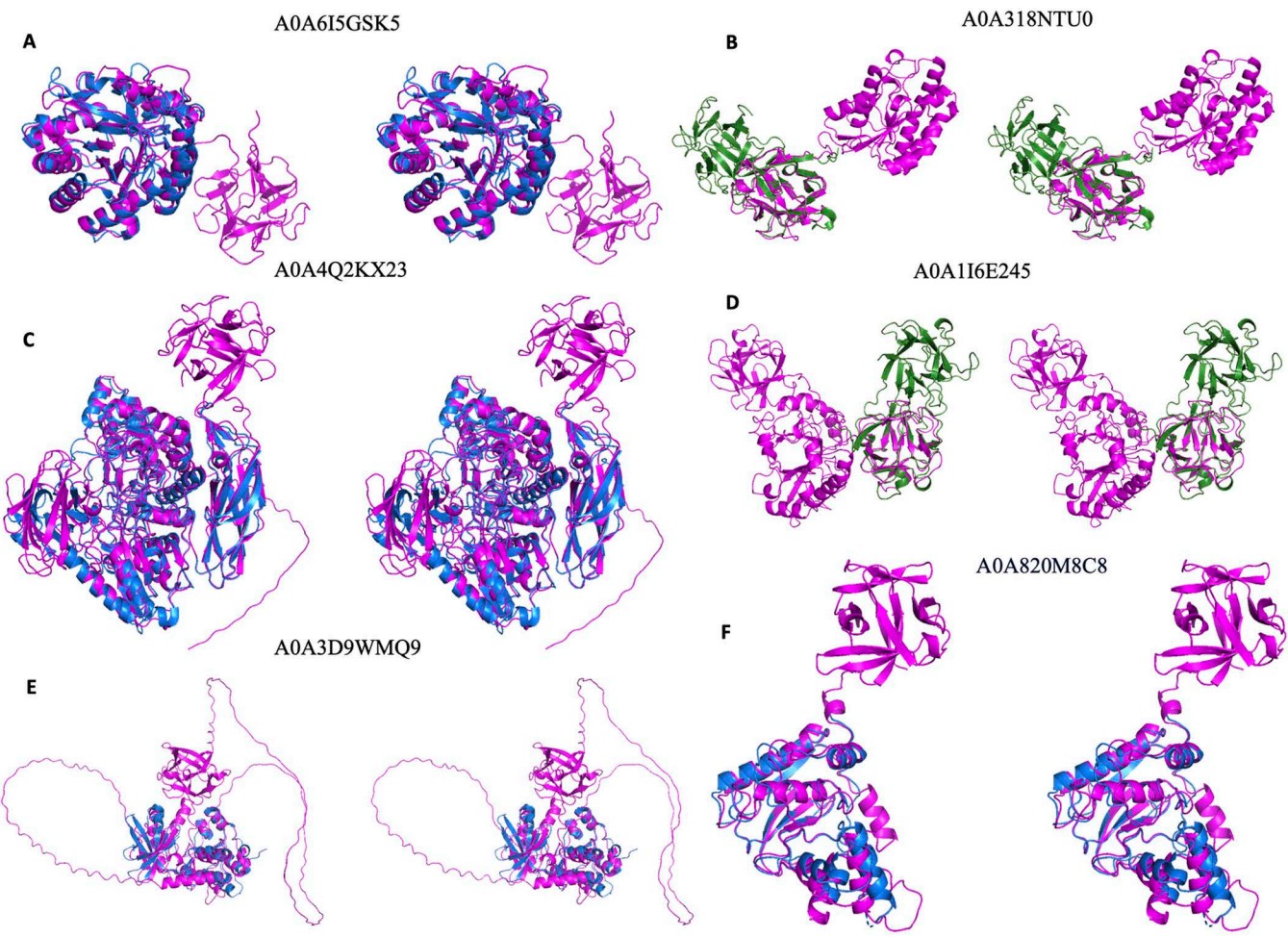

**Fig 4. Stereo diagram of the six characterized UniProt entries with predicted structures not yet represented in PDB.** PDB queries in 4b and 4d representing category-1 are in green, AFdb structures in pink, PDB hits for category-2 in 4a, 4c, 4e and 4f are in blue.

the tolerance of most positions except those that are circled as forming core region. Since there are only 9 positions out of ~150 amino acids that display low entropy with 7 positions circled as forming protein core, less than 5% of the protein has low tolerance to mutations with non-conservative substitutions. A phylogenetic tree created using the alignment of the 7 core positions, shown in Fig 6, cluster by function more than by kingdom.

## Discussion

The incomprehensible convergence in fold for billions of proteins, divergent at the sequence-level, can now be used in functional characterization of thousands of proteins under each organism using AFdb. Here, by identifying novel domain architecture in the AFdb structures containing the β-trefoil fold, we demonstrate the usefulness of fold-fold comparison in exploring functional diversity. This approach could be considered a bottom-up compared to clustering the entire structure database using fold-fold comparison, which we consider top-down. All top-down approaches including the approaches used by CATHdb, SCOP and Foldseek is challenged while clustering multidomain proteins [15,19,22]. In that domains that are highly conserved at sequence or structure-levels is likely to misplace and disperse more divergent domains into several clusters, thus posing a real challenge in functional annotation of divergent protein folds like β-trefoil.

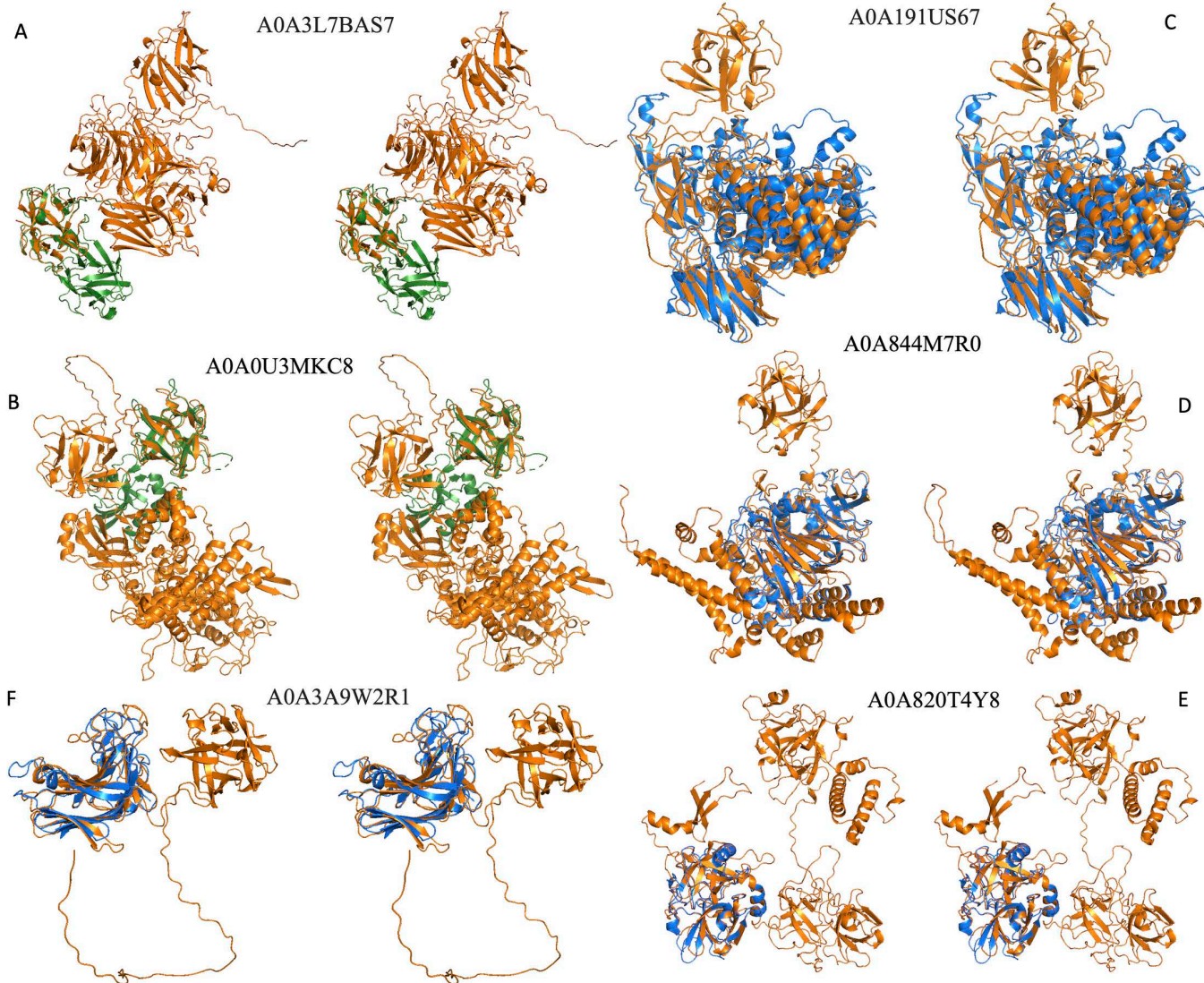

**Fig 5. Stereo diagram of the seven uncharacterized proteins from AFdb containing β-trefoil domain not part of the functions and domain architecture presented in Fig 1.** PDB queries in 5a and 5b representing category-1 are in green, AFdb structures in orange, PDB hits for categories-2 and 3 in 5c through 5f in blue.

While the SCOP and CATH database classifies β-trefoil domain containing proteins into eight (8) and four (4) distinct functions, our search in the PDB revealed nineteen (19) distinct functions for proteins containing this domain as shown in Fig 1. Except for a few, these functions span all kingdoms including plants, bacteria, fungi and Metazoa. This is in spite of the limited functional diversity in the PDB, as the database is highly redundant and/or biased towards proteins of therapeutic or economic importance to humans. Therefore, we searched the AFdb containing over 200 million predicted protein structures using representative PDB structures from the nineteen functional class as listed in Table 1. Not surprisingly, as shown in Fig 2, among the AFdb hits for a given PDB query with 80% coverage, there is a clear inverse correlation between sequence identity and r.m.s.d. for the predicted structures as expected, thus validating AlphaFold predictions. In that, AlphaFold is a mega homology modeling tool on the one end and a novel fold prediction tool on the

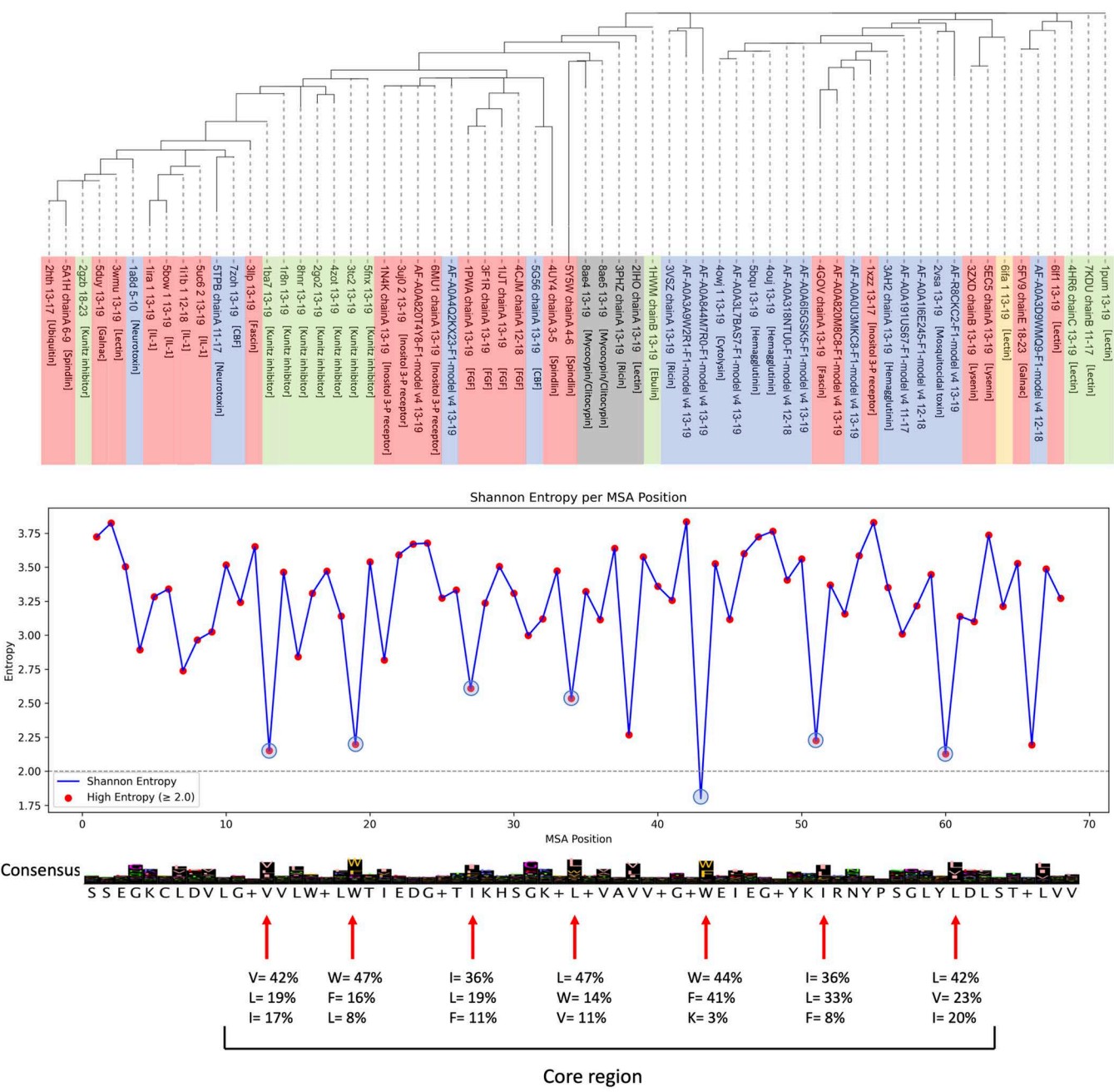

**Fig 6. Top shows phylogenetic tree from only aligning the 7 core positions with low entropy.** Bottom show Shannon entropy at positions with less than 10% gaps from a structure-based alignment. The mutation tolerance at each core position is shown with red arrows.

other. Interestingly, the steps in the plots of r.m.s.d. against sequence identity in Fig 2, represent a distinct class or order or kingdom as expected.

Among the β-trefoil containing proteins from AFdb, we discovered thirteen novel domain architectures where the β-trefoil fold is found along with an additional domain that is either not yet structurally characterized or the domain architecture is not yet represented in the PDB. With the exception of A0A820T4Y8, all the novel proteins containing a β-trefoil domain are from

culturable bacteria and are amenable to functional characterization using experimental techniques. For example, A0A3L-7BAS7 is a protein from *Micromonospora sp.* CV4, A0A191US67 is a protein from *Streptomyces parvulus*, A0A844M7R0 is a protein from *Scytonema sp.* UIC 10036 (cyanobacteria), R8CKC2 from *Bacillus cereus*, A0A820T4Y8 from *Rotaria sp. Silwood2* (Metazoa) and A0A0U3MKC8 from *Roseateles depolymerans*.

Function prediction using fold comparison is challenged by folds that may have evolved independently several times. For β-trefoil fold, reports in the literature argues that as a primordial fold, as a fold highly tolerant of mutations and as a fold that may have been built from peptides components such as β-hairpins or antibody fragments, it is most likely a divergent evolution [16,23,24]. Here, using structure-based sequence alignment of β-trefoil domains from 32 distinct functions including multiple domains we address if β-trefoil is a converging or diverging evolution. As shown in Fig 6, entropy at every position in the structure-based sequence alignment, covering less than 10% gaps among the 64 trefoil domains representing the 32 functions identified from PDB and AFdb databases, show very few positions with low-entropy, supporting a fold that is highly tolerant to mutations. A phylogenetic tree in Fig 6 generated using only the seven low-entropy positions reveal four clades mostly by function suggesting that the fold may have independently evolved at least four times, supporting convergent evolution and explaining the diversity in function, diversity in domain architecture and ability to bind to diverse substrates. If true, fold-based function prediction will hit a twilight zone for folds like β-trefoil just like sequence-based function prediction methods. At the least, our work challenges our ability to differentiate between conservation in the core amino acids stemming from divergent evolution or structural constraint imposed during convergent evolution.

## Method

### Identifying β-trefoil fold-containing proteins

As shown in Fig 7A, a few representative structures known to have β-trefoil domain and were more diverged from each other, with greater than2 Å r.m.s.d and less than 30% sequence identify, were selected from the PDB database and were used to search against the entire PDB using DALI, a structure-structure comparison tool [21]. A filter of less than 2 Å r.m.s.d and greater than 80% coverage was used to reduce false positives during this iteration. These hits were then classified both based on their annotation in the PDB and domain architecture as shown by the spokes in the outer ring in Fig 1.

The command lines used for flowchart in Fig 7A (detailed bash script of the workflow in S3 File) are:

Obtain query PDBs (β-trefoil folds from PDB)

- wget https://files.rcsb.org/download/5bow.pdb

Uploaded PDB structure to DALI server at

- https://ekhidna2.biocenter.helsinki.fi/dali

Filtering hits for r.m.s.d. threshold < 2.0 Å and alignment coverage ≥ 80% for selecting hits.

- awk 'NF > 0 && $1!~/^#/ && $1!~/^Job:/ && $1!~/^Query:/ && $4+0/ 151 * 100>= 80 && $3+0 < 2.0' dali_output_5bow.txt > dali_output_5bow_filtered.txt

As shown in Fig 7B one structure representing distinct domain architecture, discovered from the PDB search (Fig 7A), were selected to search AFdb. In order to look for known and novel functional architecture in AFdb, we interrogated hits with greater than 30% query coverage because β-trefoil domain in all the 18 representative structures ranged 30–50% of the proteins. Note that this cutoff may lead to false negatives if in novel large proteins β-trefoil domains is less than 30% of the protein. For example, proteins with domain structures like 6mu1 (S4 Fig in S1 File) where the β-trefoil region spans only 15% of the entire structure. As mentioned in the results section, the hits are binned into various categories

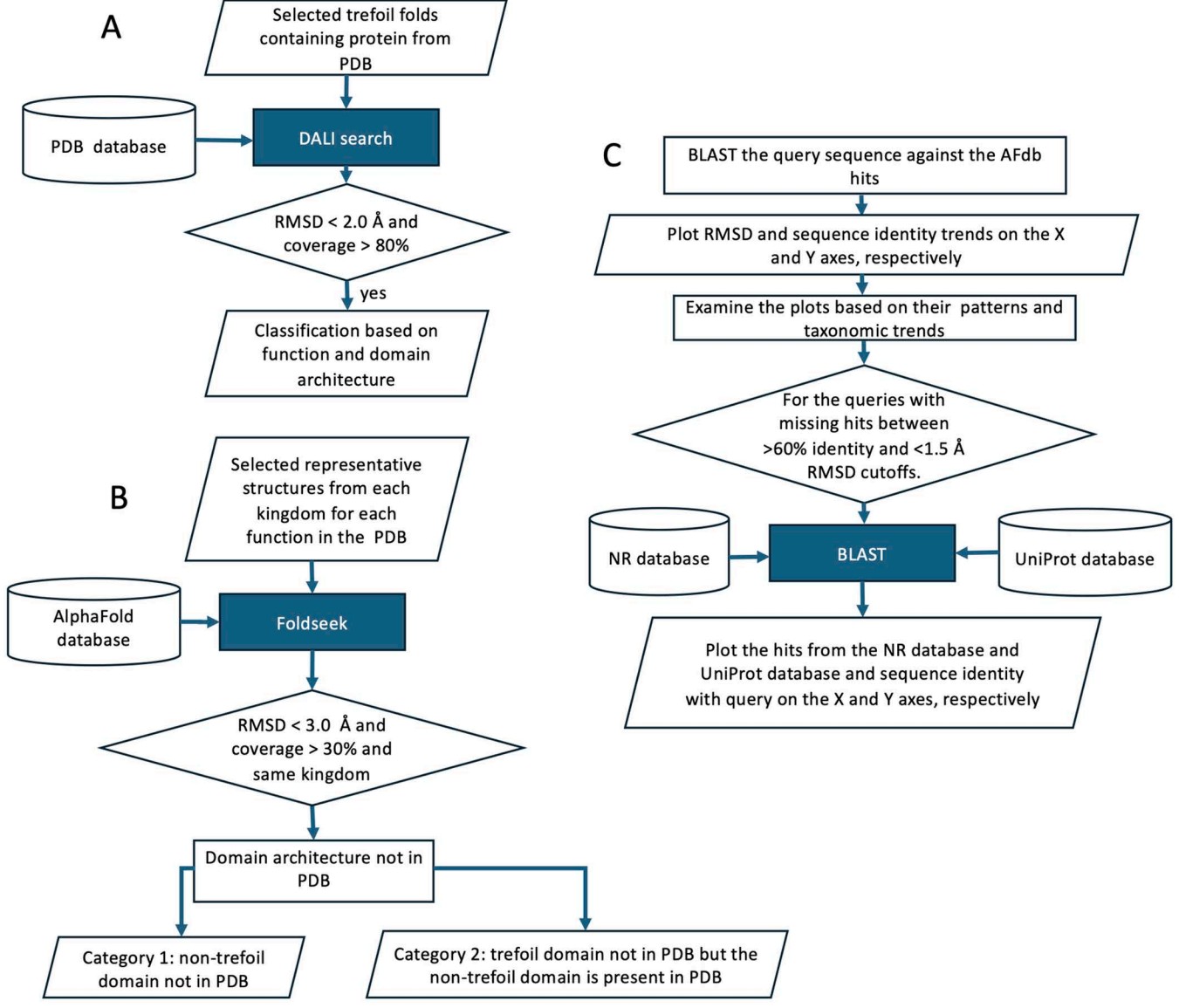

**Fig 7. Flowchart of various steps along with filtering criteria.**

depending on the presence of β-trefoil and/or other domains in PDB. The command line for pipeline in Fig 7B are follows.

- wget https://mmseqs.com/foldseek/foldseek-linux-avx2.tar.gz; tar xvzf foldseek-linux-avx2.tar.gz; export PATH = $(pwd)/foldseek/bin/:$PATH

- foldseek dataases Alphafold/UniProt up tmp_up

- foldseek easy-search query_structure1.pdb query_structure2.pdb up foldseek_results_combined_1_2.txt tmp_up --max-seqs 5000 -e 1e-4 --alignment-type 0 --format-output target,qlen,alnlen,rmsd,taxname,taxlineage

- awk –F'\t' '($4+0 < 3) && ($3+0>= 0.3 * $2) {print}' foldseek_results_combined_1_2.txt > foldseek_filtered_results.txt

AlphaFold predicts structures using multiple sequence alignment, known PDB structures, rotamer libraries, cooperative substitution, and other features using deep learning. Because of this, it is anticipated that the predicted 3D structures of proteins with large representative of homologs in sequence and structure database should be more accurate with lower r.m.s.d. for homologs with high sequence identity. The flowchart in Fig 7C shows the workflow to obtian r.m.s.d. (x-axis) against sequence identity (30%−100%, y-axis) for all the hits from AFdb against selected query structures limited to a query coverage of greater than 80% shown in Fig 2.

**Mutation tolerance analysis across 32 distinct functions for β-trefoil folds from PDB and AFdb**

To quantitatively assess the mutation tolerance of the β-trefoil fold, we performed a structure-based MSA from all 32 functions, spanning diverse taxonomic groups. For each aligned position containing less than 10% gaps, the Shannon entropy was computed to evaluate the degree of sequence variability (Python program for computing Shannon's entropy in S4 File). Positions exhibiting high entropy were indicative of considerable amino acid diversity. Only at 7 positions representing the core were significantly low, suggesting high tolerance for mutations in β-trefoil. We also generated a phylogenetic tree with only the 7 core positions shown in Fig 6 (top).

## Supporting information

**S1 File. S1 Fig.** For the PDB structures from Table 1 with gap between low r.m.s.d. (<1.5) and high sequence identity (> 50%): 1A, 2A and 3A show r.m.s.d. (x-axis) v/s Sequence identity (y-axis) plots for the AFdb Hits from 5TPB, 2IHO and 4HR6 respectively (first row), 1B and 3B show UniProt database BLAST Hits (x-axis) v/s Sequence identity (y-axis) for 5TPB and 4HR6 respectively (second row) and 1C, 2B and 3C show NR database BLAST Hits (x-axis) v/s Sequence identity (y-axis) for 5TPB, 2IHO and 4HR6 respectively (third row) . S2 Fig. Characterized hits with diverse and novel domain architecture superimposed with PDB – enlarged view of each superimposed pair as shown in Fig 3, in the same order. S3 Fig. Uncharacterized hits with diverse and novel domain architecture superimposed with PDB – enlarged view of each superimposed pair as shown in Fig 5, in the same order. S4 Fig. 6MU1 (Inositol 3-P receptor) [Note: β-trefoil domains (in this case, mostly without well-defined secondary structure elements) cover only 15% of the total structure which is as long as 2732 residues, therefore, the 30% and above coverage rule for filtering AFdb hits does not apply to this]. S5 Fig. Phylogenetic tree obtained by structure-based sequence alignment of selected β-trefoil domains (filtered to include only those positions with less than 10% gaps) for all 64 structures from PDB and AFdb. S1 Table. Novel domain architecture for proteins from AFdb containing β-trefoil fold.
(DOCX)

**S2 File.** MSA of structurally aligned sequences of 64 trefoils from 32 distinct functions filtered to include only those positions with less than 10% gaps.
(PDF)

**S3 File.** Bash-script of the pipeline workflow.
(PDF)

**S4 File.** Python program used for computing Shannon's entropy.
(PDF)

## Acknowledgments

The authors thank GoK for funding for data analysis personnel via BioIT grant and computing infrastructure via the Department of IT, BT and ST. The authors also acknowledge using resources from DBT Builder Sanction No. BT/INF/22/SP45402/2022 dated 08/03/2022 and CCB Sanction No. BT/PR40212/BTIS/137/40/2022 dated 19/12/2022.

## Author contributions

**Conceptualization:** Subhashini Srinivasan.

**Formal analysis:** Moushmi Goswami.

**Methodology:** Moushmi Goswami.

**Project administration:** Subhashini Srinivasan.

**Supervision:** Subhashini Srinivasan.

**Validation:** Moushmi Goswami.

**Visualization:** Moushmi Goswami.

**Writing – original draft:** Subhashini Srinivasan, Moushmi Goswami.

**Writing – review & editing:** Subhashini Srinivasan.

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
