## [Decision Letter · Decision Letter 0]

Dear Dr. Srinivasan,

Thank you for submitting your manuscript to PLOS ONE. After careful consideration, we feel that it has merit but does not fully meet PLOS ONE’s publication criteria as it currently stands. Therefore, we invite you to submit a revised version of the manuscript that addresses the points raised during the review process.

We look forward to receiving your revised manuscript.

Kind regards,

Sushil Mishra, Ph.D

Academic Editor

PLOS ONE

Journal Requirements:

2. In the online submission form, you indicated that [Upon request from the corresponding author].

3. Please include captions for your Supporting Information files at the end of your manuscript, and update any in-text citations to match accordingly. Please see our Supporting Information guidelines for more information: http://journals.plos.org/plosone/s/supporting-information .

Reviewers' comments:

Reviewer's Responses to Questions

**Comments to the Author**

1. Is the manuscript technically sound, and do the data support the conclusions?

Reviewer #1: Yes

Reviewer #2: Yes

2. Has the statistical analysis been performed appropriately and rigorously?

Reviewer #1: N/A

Reviewer #2: Yes

3. Have the authors made all data underlying the findings in their manuscript fully available?

Reviewer #1: No

Reviewer #2: Yes

4. Is the manuscript presented in an intelligible fashion and written in standard English?

Reviewer #1: No

Reviewer #2: Yes

Reviewer #1: The authors present an interesting investigation into the application of AlphaFold and fold-recognition software for the efficient discovery and functional annotation of novel protein folds. They argue that this approach, leveraging structural data in conjunction with sequence homology, enhances functional annotation. While the methodology is generally well-presented, several concerns require attention.

Major Concerns:

1. Code Availability: The absence of publicly available code limits the reproducibility and broader applicability of the presented methodology. Providing the code as a tool for general use would significantly enhance the study's impact.

2. Trefoil Fold Representation and Analysis: A detailed graphical representation of the trefoil fold's architectural characteristics is essential. Furthermore, a comprehensive explanation of the eighteen variant structures used in the second iteration is required. This explanation should clearly articulate their distinct features compared to the canonical trefoil domain and provide a robust justification for their selection as representative instances.

3. Mutation Tolerance Analysis: The claim that "trefoil folds appear to be highly tolerant of amino acid changing mutations" requires substantiation through a mutation survey analysis.

4. CATH Database Perspective: A thorough discussion of the CATH database's classification and perspective on the trefoil fold is necessary.

Minor Concerns:

• Language and Presentation:

o The language requires refinement to achieve a more formal and precise style.

o Sentence structure should be improved to eliminate ambiguity.

o Data presentation, including table conciseness, figure clarity, and image resolution, needs enhancement.

• "Thirty-two Functions": The mention of "thirty-two functions" lacks context and reference. Clarification is needed regarding their relevance and potential categorization.

• Wording Improvements:

o Replace the informal phrases "decades where nothing happened" and "slowly but surely" with more precise and academic language when describing the progression of homology-based structure prediction.

o The list of novel domain architectures requires clearer and more concise presentation.

• Approach Clarity: The description of the "bottom-up approach" is confusing and lacks clarity. A more precise explanation of the methodology is needed.

• Sentence Refinement: The sentence describing the improvement of structure prediction methods through multiple templates, secondary structure prediction, ab initio loop modeling, and backbone-dependent rotamer libraries requires a more streamlined and coherent structure.

In summary, while the study presents a valuable methodology, addressing these concerns, particularly regarding code availability, trefoil fold analysis, mutation tolerance, and language refinement, is essential for improving the rigor and impact of the research.

Reviewer #2: This is an interesting analysis using the beta-trefoil fold. It highlights several interesting aspects of of evolution of this fold, notably that it is likely on a trajectory of convergent evolution. The overall writing style is also fine and it is clear why they did what they did.

**Do you want your identity to be public for this peer review?** For information about this choice, including consent withdrawal, please see our Privacy Policy

Reviewer #1: **Yes: ** Neshatul Haque

Reviewer #2: No

---

## [Author Response · Author response to Decision Letter 1]

30 May 2025

Dear Reviewers,

Below find point-by-point response to the reviewers.

Reviewer's Responses to Questions

Comments to the Author

1. Is the manuscript technically sound, and do the data support the conclusions?

Reviewer #1: Yes

Reviewer #2: Yes

2. Has the statistical analysis been performed appropriately and rigorously?

Reviewer #1: N/A

Reviewer #2: Yes

3. Have the authors made all data underlying the findings in their manuscript fully available?

Reviewer #1: No

Most of the data presented are from public domain such as PDB and AFdb. We do provide accession IDs for all the proteins we identified, which are available under the respective databases. We have now provided the command-lines used in our research.

Reviewer #2: Yes

4. Is the manuscript presented in an intelligible fashion and written in standard English?

Reviewer #1: No

We have changed the flow and readability significantly..

Reviewer #2: Yes

5. Review Comments to the Author

Reviewer #1: The authors present an interesting investigation into the application of AlphaFold and fold-recognition software for the efficient discovery and functional annotation of novel protein folds. They argue that this approach, leveraging structural data in conjunction with sequence homology, enhances functional annotation. While the methodology is generally well-presented, several concerns require attention.

THANK YOU.

Major Concerns:

1. Code Availability: The absence of publicly available code limits the reproducibility and broader applicability of the presented methodology. Providing the code as a tool for general use would significantly enhance the study's impact.

We have used publicly available tools and we have now provided command lines along with options used in this work. (lines 1153-1159 and 1169-1194)

2. Trefoil Fold Representation and Analysis: A detailed graphical representation of the trefoil fold's architectural characteristics is essential. Furthermore, a comprehensive explanation of the eighteen variant structures used in the second iteration is required. This explanation should clearly articulate their distinct features compared to the canonical trefoil domain and provide a robust justification for their selection as representative instances.

● We have shown the structural arrangement of beta-hairpins, which constitutes the trefoil in the nineteen functional spoke in Figure 1.

● We have now explained it in the text (lines 333-337). It says that we selected to represent diverse domain architecture as shown in the outmost layer in Figure 1.

● We have now compared the representative structures with 5bow, which has a canonical trefoil fold, and report r.m.s.d and sequence identity within each functional spoke in Figure 1. The diversity in r.m.s.d provides the justification for selection as representative for a given function.

3. Mutation Tolerance Analysis: The claim that "trefoil folds appear to be highly tolerant of amino acid changing mutations" requires substantiation through a mutation survey analysis.

Since, a structure-based sequence alignment of diverse trefoil domains identified here represent mutations accepted during natural selection, we have computed Shannon’s entropy at each alignment position with less than 10 % gaps. The high entropy at 94% at the positions suggest a fold highly tolerant to amino acid changing mutations. Even at the positions with low entropy, multiple hydrophobic amino acids are tolerated. We have now added and whole new section titled “Interrogation of mutation tolerance in β-trefoil” this to the manuscript along with Figure 6 in lines 747-765.

4. CATH Database Perspective: A thorough discussion of the CATH database's classification and perspective on the trefoil fold is necessary.

We have now added CATH classification both in the last paragraph of the Introduction and Discussion sections in lines 219-223 and 783 respectively. We believe that the CATH database relies on annotation based on the functions of the major domains, which in majority of the cases, are more than 50% of the trefoil domain and are also more conserved at the sequence-level. Since Trefoil domain is very divergent at the sequence-level, often misses annotation based on sequence comparison. Here, we have identified many proteins from PDB and AFdb that contain at least one trefoil domain. See lines 223-228.

Minor Concerns:

• Language and Presentation:

o The language requires refinement to achieve a more formal and precise style.

o Sentence structure should be improved to eliminate ambiguity.

We have read and changed the style to make the manuscript more readable as may be clear from tracked changes throughout the manuscript.

o Data presentation, including table conciseness, figure clarity, and image resolution, needs enhancement.

We have removed the pie charts from the Table 1 and all figures are produced at high-resolutions and submitted separately for publication.

• "Thirty-two Functions": The mention of "thirty-two functions" lacks context and reference. Clarification is needed regarding their relevance and potential categorization.

Thanks for pointing this out. In line 32-36 under the Abstract we have elaborated this with context and reference.

• Wording Improvements:

We have done this throughout the document.

o Replace the informal phrases "decades where nothing happened" and "slowly but surely" with more precise and academic language when describing the progression of homology-based structure prediction.

We have replaced this in lines 155-163.

o The list of novel domain architectures requires clearer and more concise presentation.

• Approach Clarity: The description of the "bottom-up approach" is confusing and lacks clarity. A more precise explanation of the methodology is needed.

Thanks for pointing this out. Now we have added what we consider top-down to clarify why we call our approach bottom-up in lines 201-213. In the top-down approach one clusters all structures into fold/functional classes using structure-comparison similar to CATHdb clustering using sequence-based comparison. In one such applications (top-down) they report 63 million clusters based on structure comparison from AFdb. Again for multidomain proteins containing trefoil fold along with addition fold with higher sequence identify, may misplace trefoil folds into different clusters. Here we take a single fold to explore the entire database from functional classification point of view giving preference to trefoil domain and drag other domains with it for annoation

• Sentence Refinement: The sentence describing the improvement of structure prediction methods through multiple templates, secondary structure prediction, ab initio loop modeling, and backbone-dependent rotamer libraries requires a more streamlined and coherent structure.

We have modified the sentence to read better. Line 151-154.

In summary, while the study presents a valuable methodology, addressing these concerns, particularly regarding code availability, trefoil fold analysis, mutation tolerance, and language refinement, is essential for improving the rigor and impact of the research.

Thank you for your valuable comments. To our knowledge we have addressed all these concerns.

Reviewer #2: This is an interesting analysis using the beta-trefoil fold. It highlights several interesting aspects of of evolution of this fold, notably that it is likely on a trajectory of convergent evolution. The overall writing style is also fine and it is clear why they did what they did.

THANK YOU.

6. PLOS authors have the option to publish the peer review history of their article (what does this mean?). If published, this will include your full peer review and any attached files.

Do you want your identity to be public for this peer review? For information about this choice, including consent withdrawal, please see our Privacy Policy.

Reviewer #1: Yes: Neshatul Haque

Reviewer #2: No

---

## [Decision Letter · Decision Letter 1]

Tracing the function expansion for a primordial protein fold in the era of fold-based function prediction: beta-trefoil

PONE-D-25-08365R1

Dear Dr. Srinivasan,

We’re pleased to inform you that your manuscript has been judged scientifically suitable for publication and will be formally accepted for publication once it meets all outstanding technical requirements.

Kind regards,

Sushil Mishra, Ph.D

Academic Editor

PLOS ONE

Reviewers' comments:

Reviewer's Responses to Questions

**Comments to the Author**

Reviewer #1: All comments have been addressed

2. Is the manuscript technically sound, and do the data support the conclusions?

Reviewer #1: Yes

3. Has the statistical analysis been performed appropriately and rigorously?

Reviewer #1: Yes

4. Have the authors made all data underlying the findings in their manuscript fully available?

Reviewer #1: Yes

5. Is the manuscript presented in an intelligible fashion and written in standard English?

Reviewer #1: Yes

Reviewer #1: The authors have addressed all my concerns, and the manuscript is now ready for publication.

However, I couldn't locate lines 1153-1159 and 1169-1194 as referenced. While some commands are present, the code sharing could be improved by using standard formats like R, Python, or shell scripts. Although I couldn't always match the suggested changes to specific line numbers, I did confirm their implementation in the manuscript. Finally, the images in the PDF I received were unclear; hopefully, the journal has higher-resolution versions.

**Do you want your identity to be public for this peer review?** For information about this choice, including consent withdrawal, please see our Privacy Policy

Reviewer #1: **Yes: ** Neshatul Haque

---

## [Editor Report · Acceptance letter]

PONE-D-25-08365R1

PLOS ONE

Dear Dr. Srinivasan,

I'm pleased to inform you that your manuscript has been deemed suitable for publication in PLOS ONE. Congratulations! Your manuscript is now being handed over to our production team.

Kind regards,

on behalf of

Dr. Sushil Mishra

Academic Editor

PLOS ONE